# A Method to Predict CO_2_ Mass Concentration in Sheep Barns Based on the RF-PSO-LSTM Model

**DOI:** 10.3390/ani13081322

**Published:** 2023-04-12

**Authors:** Honglei Cen, Longhui Yu, Yuhai Pu, Jingbin Li, Zichen Liu, Qiang Cai, Shuangyin Liu, Jing Nie, Jianbing Ge, Jianjun Guo, Shuo Yang, Hangxing Zhao, Kang Wang

**Affiliations:** 1College of Mechanical and Electrical Engineering, Shihezi University, Shihezi 832003, China; 2Xinjiang Production and Construction Corps Key Laboratory of Modern Agricultural Machinery, Shihezi 832003, China; 3Industrial Technology Research Institute of Xinjiang Production and Construction Corps, Shihezi 832000, China; 4College of Information Science and Technology, Zhongkai University of Agriculture and Engineering, Guangzhou 510225, China

**Keywords:** CO_2_ mass concentration prediction, LSTM, PSO, random forests, sheep barn

## Abstract

**Simple Summary:**

With the change in meat sheep breeding from traditional farming to large-scale, intensified modern breeding practices, the environmental air quality in sheep barns has gradually started to receive more attention. CO_2_ concentration is an important environmental indicator in the ambient air of sheep sheds; when excess CO_2_ accumulates, it can lead to chronic hypoxia, lethargy, loss of appetite, weakness, and stress in sheep, which seriously endangers their healthy growth. Therefore, an accurate understanding of the trend of CO_2_ concentration changes in sheep housing and the precise regulation of their breeding environment are essential to ensure the welfare of sheep. Inspired by developments in deep learning technology in recent years, we propose a method to predict CO_2_ mass concentration in sheep barns based on the RF-PSO-LSTM model. The experimental results show that our proposed model has a root mean square error (RMSE) of 75.422 μg·m^−3^, a mean absolute error (MAE) of 51.839 μg·m^−3^, and a coefficient of determination (R^2^) of 0.992. The data predicted by the model are similar to the real data of a sheep barn; in fact, the prediction is better. Our proposed method can provide a reference for the prediction and regulation of ambient air quality in meat sheep barns.

**Abstract:**

In large-scale meat sheep farming, high CO_2_ concentrations in sheep sheds can lead to stress and harm the healthy growth of meat sheep, so a timely and accurate understanding of the trend of CO_2_ concentration and early regulation are essential to ensure the environmental safety of sheep sheds and the welfare of meat sheep. In order to accurately understand and regulate CO_2_ concentrations in sheep barns, we propose a prediction method based on the RF-PSO-LSTM model. The approach we propose has four main parts. First, to address the problems of data packet loss, distortion, singular values, and differences in the magnitude of the ambient air quality data collected from sheep sheds, we performed data preprocessing using mean smoothing, linear interpolation, and data normalization. Second, to address the problems of many types of ambient air quality parameters in sheep barns and possible redundancy or overlapping information, we used a random forests algorithm (RF) to screen and rank the features affecting CO_2_ mass concentration and selected the top four features (light intensity, air relative humidity, air temperature, and PM2.5 mass concentration) as the input of the model to eliminate redundant information among the variables. Then, to address the problem of manually debugging the hyperparameters of the long short-term memory model (LSTM), which is time consuming and labor intensive, as well as potentially subjective, we used a particle swarm optimization (PSO) algorithm to obtain the optimal combination of parameters, avoiding the disadvantages of selecting hyperparameters based on subjective experience. Finally, we trained the LSTM model using the optimized parameters obtained by the PSO algorithm to obtain the proposed model in this paper. The experimental results show that our proposed model has a root mean square error (RMSE) of 75.422 μg·m^−3^, a mean absolute error (MAE) of 51.839 μg·m^−3^, and a coefficient of determination (R^2^) of 0.992. The model prediction curve is close to the real curve and has a good prediction effect, which can be useful for the accurate prediction and regulation of CO_2_ concentration in sheep barns in large-scale meat sheep farming.

## 1. Introduction

China has the largest number of sheep and goats in the world and is the largest producer and consumer of meat sheep [1,2]. In order to meet the huge market demand for healthy meat sheep, transforming and upgrading meat sheep farming from the traditional free-range model to a modern model in terms of scale and intensification are inevitable [3,4]. However, the environmental air quality in sheep barns can easily deteriorate under large-scale and intensive farming, and when environmental regulation is not timely, it can threaten the normal growth and breeding of meat sheep by inducing disease outbreaks and even causing mass mortality [5,6].

Housing for sheep with a good breeding environment is the basis for disease prevention and control, considering the genetic and nutritional advantages of meat sheep [7]. Factors affecting the environmental air quality of sheep housing mainly include temperature, humidity, wind speed, and harmful gases. The gases in sheep barns mainly include O_2_, CO_2_, NH_3_, and H_2_S, among which CO_2_ is the main greenhouse gas. The CO_2_ in sheep barns is mainly produced by respiration and fecal decomposition, and its emission is influenced by the growth stage, body weight, exercise habits, and ventilation rate of the sheep [8,9,10,11]. A normal range of CO_2_ mass concentration is not harmful to the health of sheep and is not serious. When the CO_2_ mass concentration in sheep sheds is too high, the oxygen content is relatively insufficient, and sheep that live in this environment for a long time will suffer from chronic hypoxia, mental depression, loss of appetite, delayed weight gain, weakness, reduced production level, stress, and susceptibility to infectious diseases, which seriously impairs their welfare [12,13]. Therefore, it is important to study methods of predicting CO_2_ concentrations in the sheep barn breeding environments of large-scale meat sheep farms to accurately grasp the trend of CO_2_ changes and precisely regulate air quality, which is of great research value for reducing the impact of environmental stress on meat sheep growth and reproduction, preventing the occurrence of diseases and epidemics, reducing stress, and guaranteeing the welfare of sheep.

Research has been conducted on predicting CO_2_ mass concentration based on traditional machine learning methods, and some results have been obtained in predicting CO_2_ concentrations in pig houses [14], composting environments [15], and building construction environments [16,17], and with regard to urban carbon emissions [18,19], crop CO_2_ emissions [20], and ambient air pollution [21,22]. Although these CO_2_ mass concentration prediction models can express the trends of internal changes of CO_2_ in the environment and achieve certain prediction results, they require large amounts of valid data as experimental support, which creates a large and tedious workload. In addition, they can have problems, such as a long training time, slow convergence speed, susceptibility to falling into a local optimum, and poor model generalization ability, which make it difficult to meet the requirements for the timely and accurate prediction and regulation of CO_2_ mass concentration in sheep barns of large-scale meat sheep farms [23,24,25].

In recent years, with the rapid development of artificial intelligence and deep learning technology, researchers have applied deep learning techniques to a wide range of real-world problems [26,27,28,29,30,31,32,33,34,35,36,37]. Deep learning techniques have been used in crop detection [28,29], data prediction for agricultural management processes [30,31,32], crop disease detection and classification [33,34], and animal behavior recognition [35,36,37].

CO_2_ mass concentrations in sheep barns of large-scale meat sheep farms can be collected online as time series and nonlinear data, and the LSTM model, one of the typical methods of deep learning, can be used to mine future data change trends by extracting historical time series data features, which allows it to achieve certain results in time series data prediction tasks [38,39,40,41,42,43,44,45]. Wang et al. improved the LSTM model’s prediction performance by adding an adaptive attention module, which allowed the model to obtain more critical information from time series data and achieve an accurate prediction of the remaining service life of lithium–ion batteries [38]. Lin et al. first used LSTM to obtain the long time series relationship of heart rate data, then used BiLSTM to obtain the forward and backward correlation information of the data, and finally combined that with the attention mechanism to achieve an accurate prediction of heart rate [39]. Wu et al. constructed a hybrid model using a combination of LSTM and kinetic models to achieve an accurate prediction of drought occurrence [40]. Zhang et al. combined convolutional neural network (CNN) and LSTM models to construct a hybrid model (CNN-LSTM), and then combined that model with the spatiotemporal characteristics of the soil temperature field (STF) to predict the outlet temperature of energy piles [41]. Wang et al. used several machine learning methods combined with LSTM models to achieve a fast and accurate estimation of winter wheat yield over large areas based on remote sensing data [42]. In summary, considering the potential of LSTM models for complex time series data prediction tasks, in this paper we use LSTM models for the prediction of CO_2_ mass concentration in sheep barns.

To address the problem that it is difficult to predict and regulate CO_2_ mass concentrations in sheep sheds of large-scale meat sheep farms in a timely and accurate manner, we proposed a prediction method based on the RF-PSO-LSTM model. First, to address the possible problems of data packet loss, distortion, or singular values in the ambient air quality data collected online from sheep sheds, we used the mean smoothing and linear interpolation methods to repair the problematic data preprocessing and obtained a high-quality dataset. Second, to address the differences in units and magnitudes of the obtained ambient air quality data and to facilitate the study of correlations in the ambient air quality data of meat sheep barns, we normalized the data using a standardized processor. Then, to address the problem that a large variety of ambient air quality parameters as well as possible redundancies or information overlap would not only result in a complex prediction network structure, but also tend to lead to high computational complexity and low execution efficiency, we proposed to use the RF algorithm to screen and evaluate the important features of ambient CO_2_ concentration to reduce the structural complexity and computational efforts. In addition, to address the problems that the prediction results of LSTM models are susceptible to the influence of hyperparameters and the manual setting of hyperparameters is time-consuming and subjective, we proposed to use the PSO algorithm to find the optimal hyperparameter combination. Finally, the prediction results with the actual collected data were used to test the effectiveness of the method in this paper.

## 2. Materials and Methods

### 2.1. Data Source

#### 2.1.1. Test Area

The experimental area for this study was the Xinao livestock meat sheep breeding base in Lanzhouwan Town, Manas County, Xinjiang Uygur Autonomous Region (44.27° N, 86.10° E). This large-scale breeding base mainly focuses on Suffolk sheep. The total area of the meat sheep barn is 2424.31 m^2^, which includes a main area (middle, daily rest area), a shaded area (north side), and an activity area (south side), with openable and closeable passages between each area.

In summer, sheep barns are sheltered from the heat by natural ventilation and shaded areas. In winter, the main area of the sheep barn is closed for breeding, and ventilation fans are used to maintain air circulation. The test data were collected in the main area of the barn, which was a standard semi-enclosed sheep barn with an area of 442.89 m^2^ (33.3 m in length, 13.3 m in width). Sheep sheds were designed according to the Code of Management for the Construction of Livestock and Poultry Breeding Communities. The walls were made of brick and concrete, the top surface was made of steel plates, and the floor was made of mud.

The test subjects were Suffolk meat sheep, about 300 in the test barn, fed manually at regular intervals, once in the morning and once in the afternoon, with free watering and manual manure removal.

Sensors were selected to be installed under the center beam of the main area after conducting a site visit and analyzing the environment. The CO_2_ concentration sensor and total suspended particulate concentration sensor were placed 2.4 m from the ground, and the other sensors were 3.0–3.1 m from the ground; the sensor installation position is shown in Figure 1.

#### 2.1.2. Data Acquisition

In order to obtain real-time ambient air quality data of the sheep barn and ensure the consistency of samples in different seasons and time periods, ambient air quality monitoring equipment produced by Guangzhou Hairui Information Technology Co., Ltd., Guangzhou, China was used. The monitoring equipment included the following: light intensity sensor, temperature sensor, relative humidity sensor, CO_2_ concentration sensor, PM2.5 concentration sensor, PM10 concentration sensor, noise sensor, total suspended particulate (TSP) matter concentration sensor, H_2_S concentration sensor, and IoT transmission network and hubs. The response time of the monitoring equipment was less than or equal to 30 s, the repeatability was within ±2%, the linearity error was within ±2%, and the zero-point drift was within ±1%; the specific parameters are shown in Table 1.

The inter-integrated circuit (Table 1) is a serial communication protocol commonly used to connect chips to peripherals such as sensors. It was developed by Philips and is widely used for communication between various microcontrollers and digital signal processors. Modbus is a simple and widely used communication protocol used to transfer data in industrial automation systems. It is used to connect controllers or processors to other devices (e.g., sensors, actuators). The pulse width modulation (PWM) protocol is a commonly used analog signal control technique. It is typically used to convert analog signals to digital signals for reading and processing by microprocessors or other digital systems.

By using the installed monitoring equipment, we could obtain real-time ambient air quality data in the sheep barn and transmit the data to the data center of the IoT monitoring platform through communication technology.

We selected ambient air quality data from the sheep barn for the period 11 February to 25 March 2021, with a data collection interval of 10 min. The data included light intensity, temperature, relative humidity, CO_2_ mass concentration, PM2.5 mass concentration, PM10 mass concentration, noise, TSP mass concentration, and H_2_S mass concentration, with a total of 6160 sets of valid sample data. Some of the raw data collected are shown in Table 2.

#### 2.1.3. Data Preprocessing

In the process of collecting sheep barn air quality data online, there can be problems, such as external electromagnetic interference, degraded performance of aging sensors, circuit failures, and so on. These problems can lead to data deviation in the data acquisition process and packet losses in the transmission process, resulting in packet losses, distortion, or singular value problems in the collected data. To reduce the impact of these problems on prediction performance, in this study, we used the mean smoothing and linear interpolation methods to repair the problematic data preprocessing, and we obtained a high-quality dataset.

Since the obtained ambient air quality data differed in units, the data were normalized using a standardized processor in order to facilitate the study of correlations in ambient air quality data from a meat sheep barn in the winter. We preprocessed the data of the 6160 sets of valid samples and divided the preprocessed data into training, validation, and test sets in chronological order with a ratio of 7:2:1.

### 2.2. Predictive Model Construction

#### 2.2.1. Random Forest Feature Importance Ranking

The CO_2_ mass concentration in the sheep barn is influenced by the interactions among several air quality parameters, and the mechanism of interaction is complex. Due to the large variety in parameters as well as possible redundancies and information overlap, if all air quality parameters are directly input into the prediction model, it will not only result in a complex network structure, but also easily lead to low prediction accuracy, poor reliability, and high computational complexity of the model. Therefore, it is necessary to eliminate the multicollinearity among air quality parameters, screen out the features that have important effects on CO_2_ mass concentration, remove the features of lower importance, reduce the model input, optimize the model’s network structure, and improve the model’s prediction performance.

Random forests (RF) are integrated learning algorithms with decision trees as the base learners. RF not only solve the important feature-screening problem, but also have many advantages, such as simple structure, good training effects, easy implementation, and low computing cost. Given their good performance in screening important influencing features, we chose to use RF for important feature screening and evaluation of sheep barn CO_2_ mass concentration to increase the accuracy of model prediction [46].

A RF calculates feature importance mainly by calculating the average of the contributions of features above the decision tree, and then comparing the features to determine their importance. The error rate of out-of-bag (OOB) data is usually used as an evaluation indicator to aid in screening, as shown in Equation (1):(1)$${\mathrm{FIM}}_{i}=\frac{\sum \left({\mathrm{errOOB}}_{2}{-\mathrm{errOOB}}_{1}\right)}{N}$$

In Equation (1), FIM is the feature importance score, i is the number of features, N is the number of decision trees present in the random forest, errOOB_1_ is the normal out-of-bag error, and errOOB_2_ is the out-of-bag error in the presence of noise.

We used the sklearn.ensemble module in Python to call the RF function to calculate the importance of each feature.

#### 2.2.2. LSTM Model

Long short-term memory (LSTM) [47] is a further improvement on recurrent neural network (RNN). The LSTM network structure is shown in Figure 2. The LSTM model not only has the advantages of RNN in analyzing short time series, but also selects historical states that have a significant impact on the present as input by setting up a gating mechanism. This operation increases the screening of past states by the LSTM model, selectively utilizes and stores information, achieves information protection and control, and solves the problems of gradient disappearance and gradient explosion that occur in the backpropagation process of RNN training in long time sequences. Therefore, the LSTM model is more suitable for the long time series prediction problem than the RNN model.

The LSTM network structure is composed of input, hidden, and output layers. The input and hidden layers consist of a series of cyclically connected memory units. A memory unit is usually composed of one or more self-connected cells, and it also has input, output, and forgetting gates.

The LSTM model workflow has four main steps. First, the information to be forgotten in the memory cell at the previous moment is determined by multiplying the previous cell state C_t−1_ by the forgetting gate f_t_. This step uses f_t_ to remove the information in C_t−1_ that needs to be forgotten so that only the useful information is kept. Second, the new supplementary information is obtained by multiplying input gate i_t_ by the new candidate information value $\tilde{C_{t}}$, and the current cell state C_t_ can be obtained by combining the supplementary information with the reserved information; the complementary information is controlled by the input of the model and the output of the previous cell h_t−1_. Then, the C_t_ cell state values are mapped between −1 and 1 by the tanh activation function. The main purpose of using the tanh activation function is to limit the value of cell states to a controlled range to avoid problems such as gradient disappearance or gradient explosion, thus improving the stability and learning efficiency of the network. Finally, the output h_t_ of the current moment is obtained by multiplying the C_t_ cell state value by output gate o_t_; output gate o_t_ has a value between 0 and 1, which determines which information in the cell state will be output, and the output h_t_ can be used as the input h_t−1_ in the next moment or as the final output of the whole LSTM network, as shown in Equations (2)–(4):

Calculate the forgetting gate and select the information to be forgotten:(2)$$f_{t}=\sigma \left(W_{f}\cdot \left[h_{t-1}{,x}_{t}\right]{+b}_{f}\right)$$

Calculate the cell state at the current moment:(3)$$\begin{array}{l} i_{t}=\sigma \left(W_{i}\cdot \left[h_{t-1}{,x}_{t}\right]{+b}_{i}\right)\\ \tilde{C_{t}}=\tan h\left(W_{C}\cdot \left[h_{t-1}{,x}_{t}\right]{+b}_{C}\right)\\ C_{t}{=f}_{t}\cdot C_{t-1}{+i}_{t}\cdot \tilde{C_{t}} \end{array}$$

Calculate the output at the current moment:(4)$$\begin{array}{l} o_{t}=\sigma \left(W_{o}\cdot \left[h_{t-1}{,x}_{t}\right]{+b}_{o}\right)\\ h_{t}{=o}_{t}\cdot \tan h\left(C_{t}\right) \end{array}$$

In Equations (2)–(4), i_t_ is the input gate, o_t_ is the output gate, f_t_ is the forgetting gate, x_t_ is the input value at time t, and σ is the sigmoid activation function. W_i_ is the input gate weight, W_f_ is the forgetting gate weight, W_o_ is the output gate weight, W_C_ is the new candidate information weight, b_f_ is the input gate bias, b_i_ is the forgetting gate bias, b_o_ is the output gate bias, and b_C_ is the new candidate information bias. Weight W and bias b are the optimal values obtained by self-learning optimization during network training. $\tilde{C_{t}}$ is the new candidate information value at moment t, C_t_ is the cell state value at moment t, h_t−1_ is the output of the model at moment t − 1, and h_t_ is the output of the model at moment t.

#### 2.2.3. Particle Swarm Optimization

The number of neurons in each layer, the dropout probability, and the batch size hyperparameters in the LSTM model have a strong influence on the prediction results of CO_2_ mass concentration in sheep sheds. In engineering applications, the hyperparameters are usually set by personal experience or by continuous manual debugging; however, personal experience is subjective and manual debugging is time consuming and labor intensive. Therefore, we need an optimization algorithm to optimize the hyperparameters of the LSTM model.

Particle swarm optimization (PSO) [48], an intelligent optimization algorithm that uses group superposition inspired by bird flock foraging behavior, was proposed by James Kennedy and Russell Eberhart. The advantages of the PSO algorithm are its strong global search capability, the small number of parameters, the lack of gradient information, and especially the real number encoding feature, which makes it more suitable for dealing with real optimization problems.

The PSO algorithm has four main steps for solving optimization problems. First, it randomly initializes a population of particles in the solution space, each of which will have a random initial position and velocity. Second, the value of the objective function is calculated according to the location of each particle, and the globally optimal location is selected as the initial optimal solution for the search. Then, after setting the number of iterations of the particle swarm algorithm, the algorithm will enter an iterative process, updating the velocity and position of each particle at each iteration. Finally, the value of the objective function corresponding to each particle position is evaluated, the local optimal position of each particle and the global optimal position of the whole particle swarm are updated, and the algorithm iterates continuously until the end condition is met or the maximum number of iterations is reached. The algorithm flow is shown in Figure 3.

Suppose that in a D-dimensional target search space, the particle swarm number is N, the number of iterations of the current algorithm is t, and the *i*th particle state of t iterations can be represented by position and velocity vectors, as shown in Equation (5):(5)$$\begin{array}{l} x_{i}\left(t\right)=\left[x_{i,1}\left(t\right){,x}_{i,2}\left(t\right),\cdots ,x_{i,D}\left(t\right)\right]\\ v_{i}\left(t\right)=\left[v_{i,1}\left(t\right){,v}_{i,2}\left(t\right),\cdots ,v_{i,D}\left(t\right)\right] \end{array}$$

The individual optimal solution searched by the *i*th particle of current iteration t is P_best(i)(t)_, also called the local optimal solution, and the global optimal solution is G_best(t)_. In this paper, our objective was to optimize the hyperparameters of the LSTM model, in order to obtain a smaller error, which would then belong to the minimization optimization problem, as shown in Equation (6):(6)$$G_{\mathrm{best}}\left(t\right)=\min\left\{P_{\mathrm{best}\left(1\right)}\left(t\right){,P}_{\mathrm{best}\left(2\right)}\left(t\right),\cdots {,P}_{\mathrm{best}\left(N\right)}\left(t\right)\right\}$$

The velocity and position of the *i*th particle at iteration t + 1 of the algorithm can be calculated by Equation (7):(7)$$\begin{array}{l} v_{i,D}\left(t+1\right){=\omega v}_{i,D}\left(t\right){+c}_{1}r_{1}\left(P_{\mathrm{best}\left(i,D\right)}\left(t\right){-x}_{i,D}\left(t\right)\right){+c}_{2}r_{2}\left(G_{\mathrm{best}\left(D\right)}\left(t\right){-x}_{i,D}\left(t\right)\right)\\ x_{i,D}\left(t+1\right){=x}_{i,D}\left(t\right){+v}_{i,D}\left(t+1\right) \end{array}$$

In Equations (5)–(7), x is the position of the particle in the target search space; v is the velocity of the particle; t is the number of iterations; N represents the *i*th particle; N is the number of particles in the particle swarm; D is the dimension of the target search space; c_1_ and c_2_ are the learning factors, which generally take values in the range of [0, 2]; r_1_ and r_2_ are random fractional numbers in line with Bernoulli distribution and take values in the range of [0, 1]; and ω is the inertia weight factor.

The standard initialization weight method was used for LSTM model parameter setting, and the forgetting gate bias parameter was increased to prevent a large loss of information from the previous moment. The PSO algorithm uses mean absolute percentage error (MAPE), shown in Equation (8), as the fitness function to optimize the combination of the number of hidden layer neurons, dropout probability, and batch size parameters of the LSTM model.
(8)$$\mathrm{MAPE}=\frac{1}{n}\sum_{t=1}^{n}\left|\frac{A_{t}{-F}_{t}}{A_{t}}\right|$$

In Equation (8), n is the sample length of the test set, A_t_ is the true state of the CO_2_ mass concentration at time t, and F_t_ is the model prediction of the CO_2_ mass concentration at time t.

#### 2.2.4. RF-PSO-LSTM Prediction Model

In order to improve the performance of the sheep barn CO_2_ mass concentration prediction model, we proposed an organic combination of RF algorithm, PSO algorithm, and LSTM model to construct an RF-PSO-LSTM prediction model based on RF-PSO-LSTM. The method flow of this process is shown in Figure 4.

The workflow of our proposed prediction model involved four main steps. First, we remediated and standardized the ambient air quality data of the sheep barn. Second, we used RF to filter out the features with important effects on the CO_2_ mass concentration, removed the features with lower effects, reduced the input of the LSTM model, optimized the prediction model network structure, and improved the model prediction performance. Then, we optimized the number of neurons, dropout probability, and batch size hyperparameters of the LSTM model using PSO to obtain the optimal combination of hyperparameters. Finally, we used the PSO-optimized LSTM model to predict the CO_2_ mass concentration in the sheep barn.

### 2.3. Model Performance Evaluation Metrics

In order to evaluate the prediction effect and accuracy of the CO_2_ mass concentration prediction model, we selected root mean square error (RMSE), mean absolute error (MAE), and the coefficient of determination (R^2^), as shown in Equations (9)–(11):(9)$$\mathrm{RMSE}=\sqrt{\frac{1}{N}\sum_{i=1}^{N}{\left(y_{i}-\overset{\wedge }{y_{i}}\right)}^{2}}$$
(10)$$\mathrm{MAE}=\frac{1}{N}\sum_{i=1}^{N}\left|y_{i}-\overset{\wedge }{y_{i}}\right|$$
(11)$$R^{2}=1-\left(\sum_{i=1}^{N}{\left(y_{i}-\overset{\wedge }{y_{i}}\right)}^{2}/\sum_{i=1}^{N}{\left(y_{i}-\overset{-}{y_{i}}\right)}^{2}\right)$$

In Equations (9)–(11), $\overset{}{y_{i}}$ is the actual value, $\overset{\wedge }{y_{i}}$ is the predicted value, $\overset{-}{y_{i}}$ is the mean value of the dependent variable in the test set, and N is the number of samples in the test set.

The smaller the values of RMSE and MAE, the smaller the error between the predicted and actual values of the model and the higher the accuracy. The value of R^2^ ranges from 0 to 1, and the closer the value is to 1, the better the reliability of the model prediction results.

### 2.4. Model Test Platform

In this paper, the models were trained, validated, and tested on the same computer. The computer configuration was based on a Windows 10 operating system with NVIDIA GeForce RTX3060 GPU and AMD Ryzen7 5800H CPU@3.20 GHz processor, running on 16 G of RAM. All models in this study were built using Python, with the LSTM model based on the TensorFlow deep learning framework and PyCharm development tool.

## 3. Results and Discussion

### 3.1. PSO Algorithm Parameter Setting

The parameters of the PSO algorithm in this paper were set as shown in Table 3. The LSTM model used the mean square error (MSE) loss function to calculate the loss value of the model during training, and the optimizer was Adam. The number of neurons, dropout probability, and batch size hyperparameters of the LSTM model were obtained by the PSO algorithm for finding the best results.

### 3.2. Determination of LSTM Model Structure

The structure of the model needed to be determined when predicting time series using the LSTM model, consisting of input, hidden, and output layers. The input layer is the key to the data transfer of the model and is the first part of the whole model to make predictions. The output layer outputs the results of the model’s prediction of CO_2_ mass concentration and is the final link in the overall model to make predictions. LSTM models usually have only one input layer and one output layer. The structure of the LSTM model mainly lies in the different hidden layers; therefore, we need to determine the number of hidden layers in the model.

With a small number of hidden layers, the model may not be able to fully learn the relationships in the data, and the fitting ability of the data may be insufficient; as a result, the model may not achieve the expected results for CO_2_ mass concentration prediction. Too many hidden layers will lead to overfitting the model to the data, resulting in a poor generalization ability; additionally, the model will have more parameters and a complex structure. In summary, we need to set the number of hidden layers reasonably.

In this experiment, we selected the number of hidden layers as 1–5, and we used the RMSE, MAE, R^2^, and model parameters as the evaluation metrics, with a time step set to 20 by default. The test results are shown in Table 4.

From Table 4, we can see that the model has the lowest number of parameters when there is one hidden layer. When there are five hidden layers, the model has the highest number of model parameters.

With one hidden layer, the RMSE of the model was 123.959 μg·m^−3^, the MAE was 95.315 μg·m^−3^, the R^2^ was 0.978, and the parameter size was 32,251. In this case, the model may not be able to fully learn the complex action relationships in the data due to the low number of hidden layers.

With two hidden layers, the RMSE was 108.177 μg·m^−3^, MAE was 83.187 μg·m^−3^, R^2^ was 0.984, parameter size was 52,451, and the index values of the model were optimal, between one and five hidden layers.

Compared with the model with one hidden layer, the RMSE of the model with two hidden layers decreased by 15.782 μg·m^−3^, the MAE decreased by 12.128 μg·m^−3^, and the R^2^ increased by 0.006, showing that the model with two hidden layers could adequately learn the connections in the data and have fewer errors in predicting the CO_2_ mass concentration. Models with three to five hidden layers may have a relatively complex structure due to more parameters, which leads to larger errors in the prediction results.

The experiments show that the model with two hidden layers had a better prediction effect, less structural complexity, less computation, faster training, and faster running speed. Thus, the model structure with two hidden layers was chosen in this study.

### 3.3. Optimal Time Step

We used the LSTM model for time series prediction, which requires feature acquisition by a time step. The time step is a very important parameter for LSTM models because it determines the size of the feature composition structure and the amount of data required for the model during training, validation, and testing. The size of the time step directly affects the performance of model training and prediction; thus, we needed to set a reasonable value for this parameter to ensure good model performance.

In our experiments, we used the grid search method for time steps T ∈ {1, 20, 40, 60, 80, 100} [49,50,51]. We used the RMSE, MAE, and R^2^ as the evaluation indexes of the model to filter the optimal time step. The experimental results are shown in Table 5.

The values of the performance metrics of the LSTM model when the time step is set to T ∈ {1, 20, 40, 60, 80, 100} are shown in Table 5. When the range of values for the time step T are ∈ {1, 20}, it can be seen that the model performs better with T = 20 than with T = 1. The reason for this is that the feature datum produced by time step T = 1 is one feature data point, which is not the same as time series data over a time span and cannot express the relationship between continuous feature data points.

When the value range of the time step is T ∈ {20, 40, 60}, it can be seen that the prediction error of the LSTM model gradually decreases and the performance gradually improves. The LSTM model with T = 60 has the best performance because the feature data produced by this time step can better represent the relationship between continuous feature data points.

When the time step is T ∈ {60, 80, 100}, it can be seen that the overall prediction error of the LSTM model increases and the performance decreases due to the increased time step. Further, if the time step is larger, fewer data points are used for training; thus, we believe that the model is not sufficiently trained, which is the same as the conclusion reached in [51].

In summary, the prediction error of the LSTM model is the lowest and the performance is the best when time step T = 60. Therefore, in this paper, the time step of the model was chosen as T = 60, and subsequent experiments were conducted on this basis.

### 3.4. Feature Importance Ranking and Filtering

We collected a total of nine categories of environmental quality parameters in the sheep barn using the IoT. CO_2_ mass concentration is influenced by a variety of parameters, some of which show a strong correlation to it, and these parameters are called important features.

To filter the important features, we used the RF algorithm to calculate eight parameters to obtain their degree of importance and rank them in the following order: light intensity, air relative humidity, air temperature, PM2.5 mass concentration, PM10 mass concentration, noise, TSP mass concentration, and H_2_S mass concentration; the scores are shown in Table 6.

We selected different numbers of participants to input into the model in order to verify the effectiveness of the RF algorithm in order of ranking for the experiment, and we obtained the MAE variation curve as shown in Figure 5.

As seen in Figure 5, with one feature parameter, although the input dimension of the model is the smallest, the model has a poor fit and the MAE is the largest. With three feature parameters, although the input dimension of the model is smaller, the model is not fully developed, the fitting effect is average, and the MAE can be further reduced.

With four feature parameters, the MAE of the model is further reduced, the model fits better, and the input dimension is smaller. With five to eight feature parameters, the MAE is not much different from that of the model with four feature parameters, and the fitting effect is similar, but the input dimension increases.

In summary, it can be seen that with four feature parameters, the model can develop fully, the average absolute error is relatively low, the fitting effect is more satisfactory, and the input dimension is more reasonable.

In order to reduce the input dimension, optimize the network structure, and reduce the computational complexity of the model, we selected the top four parameters (light intensity, air relative humidity, air temperature, and PM2.5 mass concentration) as the prediction model inputs.

### 3.5. PSO Results for Hyperparameter Search

After determining the LSTM model structure, optimal step size, and important features, we used the PSO algorithm to find the optimal number of neurons, dropout probability, and batch size hyperparameters for the LSTM model. The results of the PSO algorithm were as follows: there were 64 neurons in the input layer, 128 neurons in hidden layer 1, 32 neurons in hidden layer 2, a dropout probability of 0.1, and a batch size of 32. We needed to train the LSTM model after determining the hyperparameters, and the training loss value changes are shown in Figure 6.

Figure 6 shows that the initial values for the training loss and validation loss of the model were 0.0213 and 0.0057, respectively. Although the initial loss of the model was high, the value rapidly decreased as the training proceeded, because the model updates the weights during the backpropagation process, gradually improving the fitting ability.

When the network training exceeded 260 epochs, the training loss value gradually stabilized between 0.0008 and 0.0009, and the validation loss value gradually stabilized between 0.0006 and 0.0007. The convergence of the loss values of the overall model shows only slight oscillations, indicating the completion of network model training.

The final RF-PSO-LSTM model in this paper was obtained after the model training was completed; Figure 7 shows the prediction effect of the model for CO_2_ mass concentration in a sheep barn. As can be seen in Figure 7a, the overall trend of our model’s prediction of CO_2_ mass concentration was similar to the actual CO_2_ mass concentration. Our model predicted the peak at the same point at which the sheep house CO_2_ mass concentration reached the peak. This demonstrates the ability of our proposed model to act as an early warning when the CO_2_ mass concentration in a sheep barn reaches a certain level, safeguarding the welfare of meat sheep to some extent.

Figure 7b further shows the difference between the CO_2_ mass concentration predicted by our model and the actual CO_2_ mass concentration in the sheep barn. It can be seen that although the predicted concentration of the model is very similar to the actual concentration, it is not as smooth as the actual value in terms of data smoothing. We determined that this could be related to changes in the environmental parameters of the sheep barn, which is a relatively slow process, and the interactions between parameters are slow, so the changes in CO_2_ mass concentration in the sheep barn are relatively smooth.

### 3.6. Comparative Analysis of Hyperparameter Predictions

In order to verify the effectiveness of the hyperparameter search results of the PSO algorithm for the LSTM model, we set different hyperparameters for the LSTM model for comparison tests. The model evaluation metrics were the RMSE, MAE, and R^2^, and the experimental results are shown in Table 7.

To determine the effectiveness of the PSO algorithm for the batch size hyperparameter search of the LSTM model, we only changed the batch size of the model, and set the RF-LSTM_1 model to have a batch size of 64 and the RF-LSTM_2 model to have a batch size of 128. It can be seen from the table that the RF-PSO-LSTM model with a batch size of 32 has the lowest RMSE and MAE values, indicating that this model has the least error in predicting the CO_2_ mass concentration in the sheep shed.

To determine the effectiveness of the PSO algorithm for the dropout hyperparameter search of the LSTM model, we changed only the magnitude of the dropout value and set the RF-LSTM_3 model with a dropout value of 0.2 and the RF-LSTM_4 model with a dropout value of 0.3. The table shows that as the dropout value increased, the RF-LSTM_4 model had a larger prediction error than the RF-LSTM_3 model in the prediction of CO_2_ mass concentration in the sheep shed. The RF-PSO-LSTM model with a dropout value of 0.1 predicted the CO_2_ mass concentration with fewer errors and better results.

To determine the effectiveness of the PSO algorithm for the hyperparametric optimization of neurons in the LSTM model, we experimented by changing only the number of neurons in the input layer, hidden layer 1, and hidden layer 2 of the model. Specifically, we set up the RF-LSTM_5–11 models for comparison with the RF-PSO-LSTM model proposed in this paper. We found from the test indicators in the table that more neurons in the model is not better, and likewise, fewer is not better either. The number of neurons in each layer of the model needs to be reasonably configured to maximize the performance of the model.

### 3.7. Comparative Analysis of Model Predictions

In order to verify the difference between our proposed RF-PSO-LSTM model and other models in predicting the CO_2_ mass concentration in sheep sheds, we used the gradient boosting regression tree (GBRT) algorithm, the light gradient-boosting machine (LightGBM) algorithm, the support vector regression (SVR) algorithm, and the random forest regression (RFR) algorithm models for the experimental analysis. The results are shown in Table 8.

The RFR, SVR, GBRT, and LightGBM models were obtained by training with all the features. The RF-RFR, RF-SVR, RF-GBRT, and RF-LightGBM models were obtained by training with the four features filtered by the RF algorithm.

It can be seen from Table 8 that compared with that of the RFR model, the RMSE of the RF-RFR model increased by 4.471 μg·m^−3^, the MAE decreased by 6.861 μg·m^−3^, and the R^2^ decreased by 0.002. Compared with the SVR model, the RMSE of the RF-SVR model decreased by 43.488 μg·m^−3^, the MAE decreased by 43.174 μg·m^−3^, and the R^2^ increased by 0.065. Compared with the GBRT model, the RMSE of the RF-GBRT model decreased by 4.475 μg·m^−3^, the MAE decreased by 2.176 μg·m^−3^, and the R^2^ increased by 0.004. Compared with the LightGBM model, the RMSE of the RF-LightGBM model decreased by 8.332 μg·m^−3^, the MAE decreased by 7.127 μg·m^−3^, and the R^2^ increased by 0.006.

We found that the model that first uses the RF algorithm to filter the important features and then trains using the filtered features has a lower MAE value and a higher R^2^ value than the model that trains using all features. The low RMSE and MAE of the model indicate that it has a small error in predicting CO_2_ mass concentration. The high R^2^ value of the model indicates that it has a high reliability in predicting CO_2_ mass concentration.

Among the compared models, the RF-RFR model predicted an RMSE of 220.844 μg·m^−3^, an MAE of 138.994 μg·m^−3^, and an R^2^ of 0.937 for the CO_2_ mass concentration in sheep sheds, which are the best predicted results.

The differences between our proposed model and the RF-RFR model can be seen in the table. Specifically, compared with RF-RFR, the RMSE of our model decreased by 145.422 μg·m^−3^, the MAE decreased by 87.155 μg·m^−3^, and the R^2^ increased by 0.055. Our proposed model had a better performance than the other models in predicting the CO_2_ mass concentration in sheep barns.

In summary, the RF-PSO-LSTM prediction model has a higher accuracy and a better fit, which are beneficial for single time series prediction with better real-time performance. Our model can be used for predicting sheep barn CO_2_ mass concentrations at large-scale meat sheep farms, providing a strong decision basis for early warning while improving the welfare of sheep.

## 4. Conclusions

In precision animal husbandry, the accurate prediction and early warnings of CO_2_ mass concentrations in large-scale sheep barns are an important research hotspot. The research work in this paper provides a reference for such predictions and early warnings, and the following conclusions were obtained:(1)The RF algorithm was able to filter out the important features affecting the prediction of CO_2_ mass concentration in sheep barns and remove features of lower importance, reducing the input to the model and the complexity of the data. The experimental results show that training the model using the filtered important features can improve the prediction performance.(2)We used the PSO algorithm to find the optimal number of neurons, dropout value, and batch size hyperparameters of the LSTM model and obtain the optimal combination of hyperparameters, avoiding the disadvantages of manual selection of hyperparameters.(3)The experimental results show that our proposed RF-PSO-LSTM model could effectively predict the trend of CO_2_ mass concentration in sheep sheds with a higher accuracy than typical prediction models such as RFR, SVR, GBRT, and LightGBM. The prediction results of our model can provide important support for improving the growing environment of meat sheep, which is conducive to improving the welfare of the sheep.

In short, we hope that our model can provide some help and reference for air quality improvements and the prediction of CO_2_ mass concentration in sheep barns at large-scale meat sheep farms.

## Figures and Tables

**Figure 1 animals-13-01322-f001:**
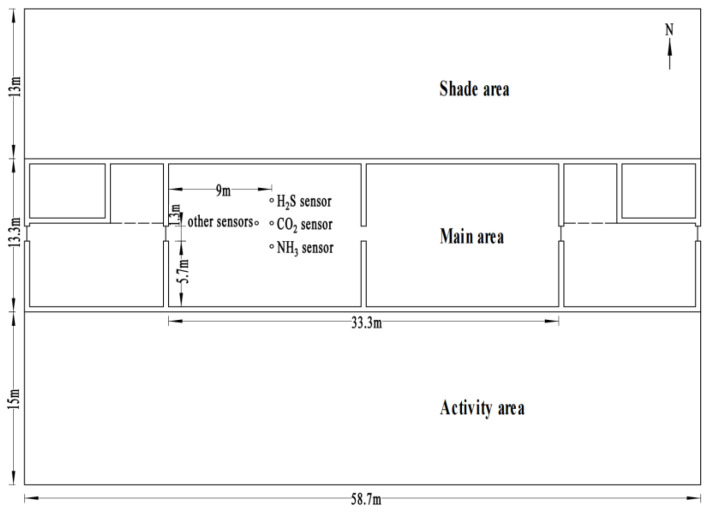
Schematic diagram of sensor installation position.

**Figure 2 animals-13-01322-f002:**
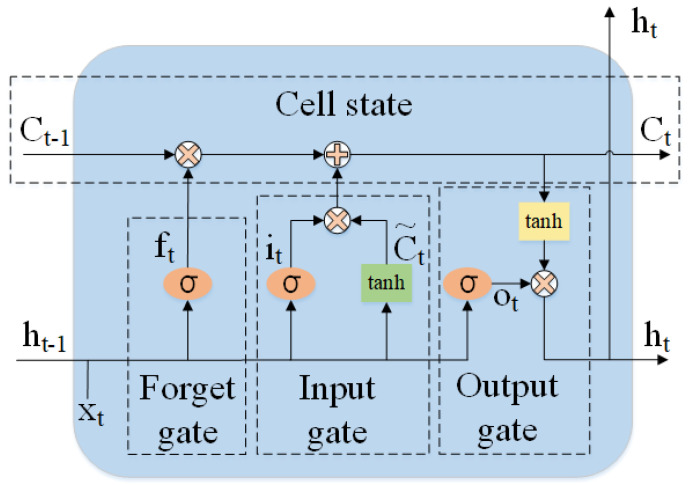
Schematic diagram of LSTM network structure.

**Figure 3 animals-13-01322-f003:**
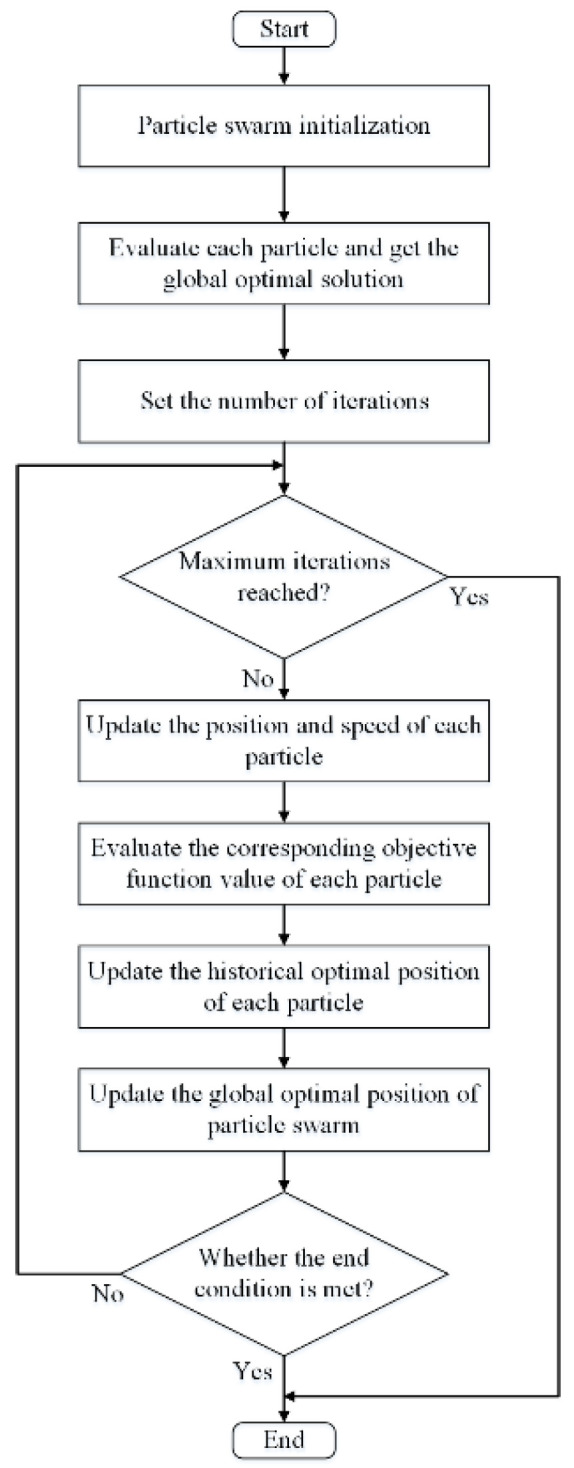
Flowchart of PSO algorithm.

**Figure 4 animals-13-01322-f004:**
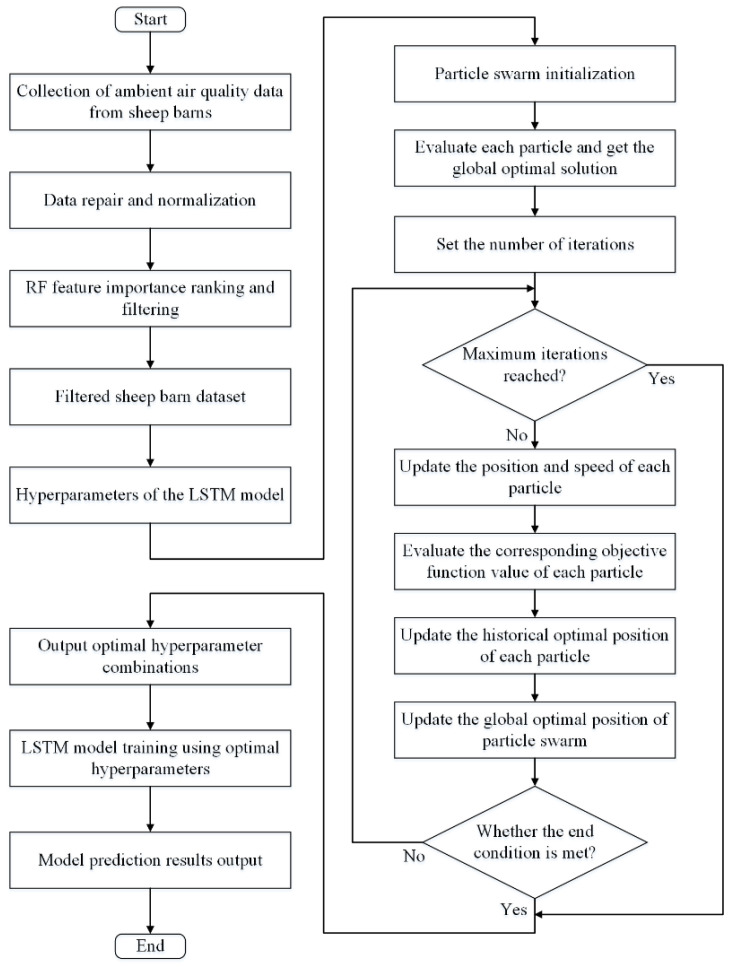
Flowchart of RF-PSO-LSTM prediction model.

**Figure 5 animals-13-01322-f005:**
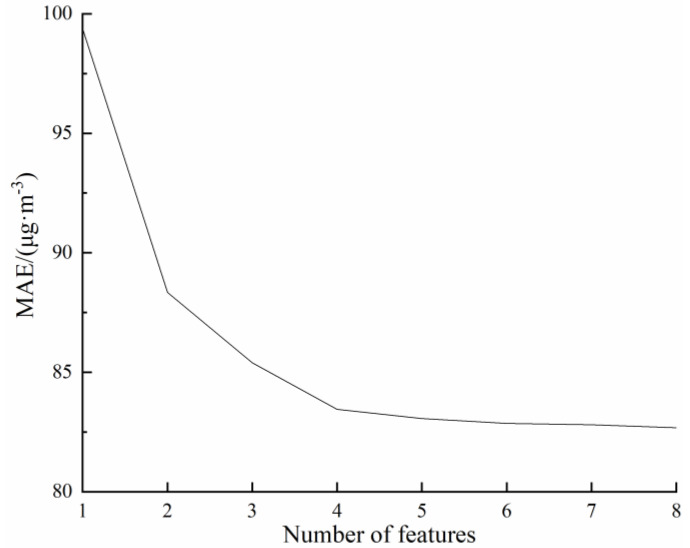
Average absolute error curve based on number of features.

**Figure 6 animals-13-01322-f006:**
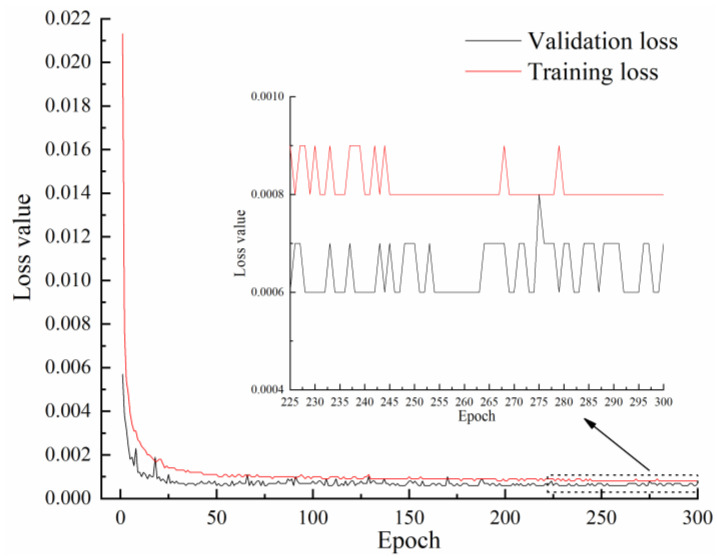
Change of loss value during training.

**Figure 7 animals-13-01322-f007:**
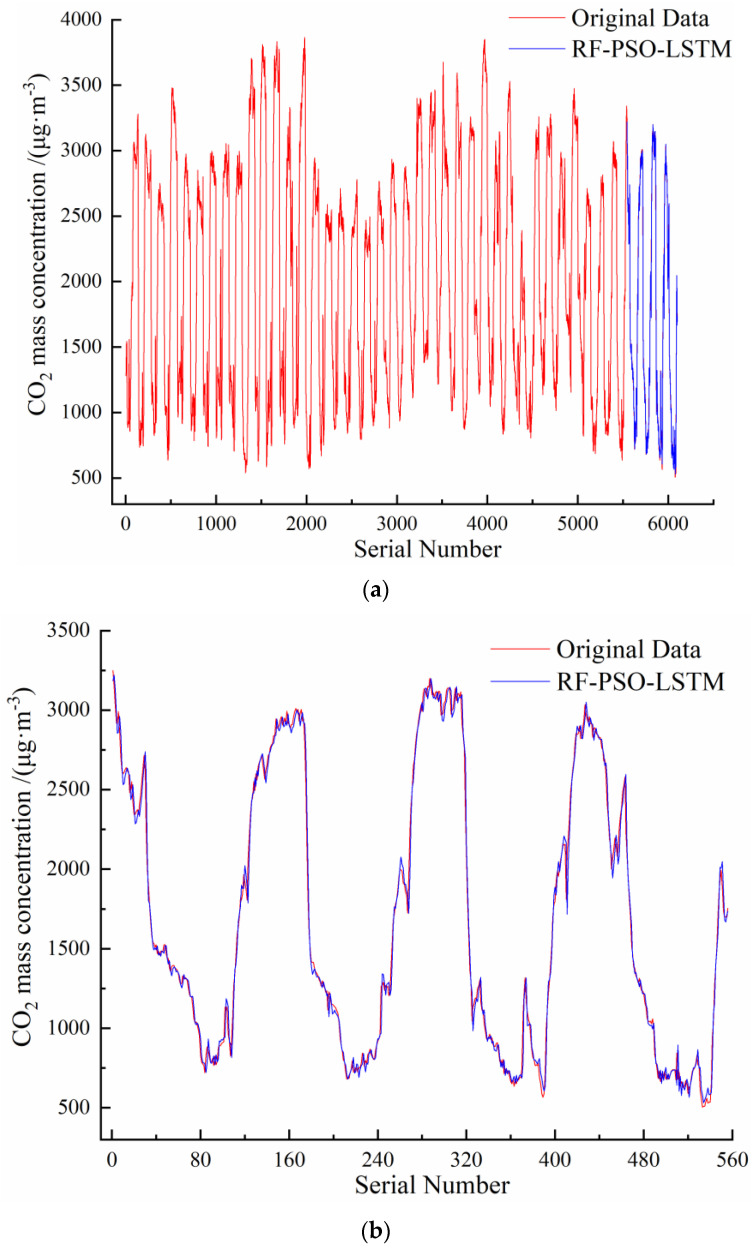
Prediction effect of RF-PSO-LSTM model: (**a**) overall forecast trend; (**b**) model prediction.

**Table 1 animals-13-01322-t001:** Technical parameters of sensors.

Testing Index	Measurement Range	Accuracy	Agreement
Light intensity (lx)	0~65,535	±5	IIC
Air temperature (°C)	−40~105	±0.4	IIC
Air relative humidity (%)	0~100	±5	IIC
Noise (dB)	30~120	±5	IIC
PM2.5 mass concentration (μg·m^−3^)	0~999.9	±7%	Modbus
PM10 mass concentration (μg·m^−3^)	0~999.9	±7%	Modbus
CO_2_ mass concentration (μg·m^−3^)	0~50,000	±50	PWM
TSP mass concentration (μg·m^−3^)	0~999.9	±7%	PWM
H_2_S mass concentration (μg·m^−3^)	0~10	±3%	PWM

**Table 2 animals-13-01322-t002:** Selected raw data of ambient air quality of sheep barn.

Testing Index	11 February 2021	11 February 2021	11 February 2021	11 February 2021	11 February 2021	11 February 2021
	10:12:18	10:22:10	10:32:09	10:42:10	10:52:23	11:02:16
Light intensity (lx)	24	30	39	42	97	122
Air temperature (°C)	1.5	1.5	1.5	1.6	1.6	1.7
Air relative humidity (%)	85.7	86.1	86.4	86.6	86.8	87.1
Noise (dB)	32	80.8	59.4	69.8	32	45.3
PM2.5 mass concentration (μg·m^−3^)	13.4	14.2	12.4	12.9	12.3	11.9
PM10 mass concentration (μg·m^−3^)	48.2	38.2	42.1	36.4	24.9	34.1
CO_2_ mass concentration (μg·m^−3^)	1300	1285	1315	1330	1420	1425
TSP mass concentration (μg·m^−3^)	76.3	65	67.5	61.1	46.2	57
H_2_S mass concentration (μg·m^−3^)	8.4	8.4	8.2	8.4	8.4	8.4

**Table 3 animals-13-01322-t003:** PSO algorithm parameter settings.

Parameter	Value
Inertia weighting factor w	0.5
Learning factor c_1_	1.3
Learning factor c_2_	1.4
Search for spatial dimension D	3
r_1_	0.6
r_2_	0.8
Number of particles N	50
Number of iterations	100

**Table 4 animals-13-01322-t004:** Indicators for number of hidden layers in the model.

Number of Hidden Layers	RMSE (μg·m^−3^)	MAE (μg·m^−3^)	R^2^	Model Parameters
1	123.959	95.315	0.978	32,251
2	108.177	83.187	0.984	52,451
3	127.123	97.337	0.975	72,651
4	143.066	109.88	0.972	92,851
5	165.080	125.849	0.959	113,051

**Table 5 animals-13-01322-t005:** Indicators at different time steps.

Time Step	RMSE (μg·m^−3^)	MAE (μg·m^−3^)	R^2^
1	108.217	85.161	0.981
20	108.177	83.187	0.984
40	111.135	82.41	0.983
60	109.586	81.8	0.982
80	119.726	87.69	0.979
100	123.212	90.919	0.978

**Table 6 animals-13-01322-t006:** Eight parameter features ranked by importance.

Order of Importance	Parameter	Importance Score
1	Light intensity (lx)	0.750228
2	Air relative humidity (%)	0.114946
3	Air temperature (°C)	0.056363
4	PM2.5 mass concentration (μg·m^−3^)	0.027768
5	PM10 mass concentration (μg·m^−3^)	0.018287
6	Noise (dB)	0.013143
7	TSP mass concentration (μg·m^−3^)	0.011485
8	H_2_S mass concentration (μg·m^−3^)	0.007780

**Table 7 animals-13-01322-t007:** Prediction results of LSTM model with different hyperparameters.

Model Name	Number of Neurons in Input Layer	Number of Neurons in Hidden Layer 1	Number of Neurons in Hidden Layer 2	Dropout	Batch Size	RMSE (μg·m^−3^)	MAE (μg·m^−3^)	R^2^
RF-PSO-LSTM	64	128	32	0.1	32	75.422	51.839	0.992
RF-LSTM_1	64	128	32	0.1	64	79.321	54.592	0.991
RF-LSTM_2	64	128	32	0.1	128	79.065	57.540	0.991
RF-LSTM_3	64	128	32	0.2	32	78.503	54.267	0.991
RF-LSTM_4	64	128	32	0.3	32	78.864	55.039	0.991
RF-LSTM_5	64	128	64	0.1	32	78.230	54.361	0.991
RF-LSTM_6	64	128	128	0.1	32	77.077	52.956	0.991
RF-LSTM_7	64	64	32	0.1	32	78.250	54.089	0.991
RF-LSTM_8	64	256	32	0.1	32	77.180	53.072	0.991
RF-LSTM_9	32	128	32	0.1	32	78.714	53.785	0.991
RF-LSTM_10	128	128	32	0.1	32	79.167	54.053	0.991
RF-LSTM_11	256	128	32	0.1	32	77.710	52.995	0.991

**Table 8 animals-13-01322-t008:** Comparison of prediction performance of different models.

Model	RMSE (μg·m^−3^)	MAE (μg·m^−3^)	R^2^
RFR	216.373	145.855	0.939
SVR	589.336	484.475	0.545
GBRT	285.102	213.499	0.895
LightGBM	288.001	209.788	0.891
RF-RFR	220.844	138.994	0.937
RF-SVR	545.848	441.301	0.610
RF-GBRT	280.627	211.323	0.899
RF-LightGBM	279.669	202.661	0.897
RF-PSO-LSTM	75.422	51.839	0.992

## Data Availability

The data presented in this study are available on request from the corresponding author. The data are not publicly available because they are part of an ongoing study.

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
