# Peer review of "A Method to Predict CO2 Mass Concentration in Sheep Barns Based on the RF-PSO-LSTM Model"

_animals, 2023, doi:10.3390/ani13081322_

Round 1

Reviewer 1 Report

I suggest you to use impersonal sentences in your article. So sentences like "Finally, we compared the prediction results.." (line 143) should be rewritten as "Finally, the prediction results  ... were compared".

Table 2. I suggest you to show collected data that not randomly chosen, but, for example, mean values for weeks or monthes. 

Author Response

Dear Reviewer:

Thanks very much for taking your time to review our manuscript entitled “A method to predict CO2 mass concentration in sheep barn based on RF-PSO-LSTM model”(ID: animals-2233884).

We appreciate all your comments and suggestions. Those comments are all valuable and very helpful for revising and improving our paper, as well as the important guiding significance to our research. We have studied the comments carefully and have made the corrections which we hope meet with approval. We have responded to your suggestions as follows:

Point 1: I suggest you to use impersonal sentences in your article. So sentences like "Finally, we compared the prediction results.." (line 143) should be rewritten as "Finally, the prediction results  ... were compared".

Response 1: We are very grateful for the suggestions you have made. We have given your suggestions careful thought.

We strongly agree with you that avoiding emotional sentences in an article can make it more reasonable. At the same time, we think that using a few sentences with emotional overtones in the text may increase the readability of the text and improve the reading experience.

We apologize that our answer may not be very comprehensive. We thank you again for your suggestions.

We have made changes based on your suggestions, which can be seen in the paper in the 150, 180, 527, and so on line.

Point 2: Table 2. I suggest you to show collected data that not randomly chosen, but, for example, mean values for weeks or monthes.

Response 2: We are very grateful for the suggestions you have made. We have given your suggestions careful thought.

We agree with you that showing data averaged over several weeks or months can help us better understand the data from the sheep barn. At the same time, we feel that a random display of the data may allow the reader to observe the true fluctuation of the data changes and improve the reading experience.

Our answer may not be very comprehensive and may have caused you some misunderstanding, we express our sincere apologies to you. We thank you again for your suggestions.

We once again express our heartfelt thanks to you.

Reviewer 2 Report

This article proposed a method to predict CO2 concentration using the RF-PSO-LSTM model. The analysis seems helpful for researchers to model air quality in a similar area. I have some comments for your consideration.

I have a question regarding the need for this specific barn. Ventilation rate seems to be one of the most impactful factors in the CO2 concentration inside the barn. However, that was not included in the testing index (input variable). Sometimes it is very difficult to obtain continuous ventilation rate data, but is there a way to somehow roughly correlate ventilation rate with one of the existing testing indexes such as air temperature? Ventilation rates highly depend on temperature. As you mentioned, in summer, the barn uses natural ventilation, and in winter, it uses mechanical ventilation. If you can provide some analysis on ventilation (quantitatively or qualitatively), that would greatly enhance the application of this model.

In Line 68, I would not call CO2 a “harmful gas”.

You mentioned that the height of the sensors is 2.4 and 3.0-3.1 meters from the ground. How tall is the ceiling of the barn? The barn has a decent size. Regarding your measurement equipment in the paragraph of line 172, where was the location of the equipment during sampling? How did you control the spatial variability of data with just 1 sampling location instead of multiple?

The number of significant figures depends on the accuracy of the measurement equipment. The result of your model output (Table 4) seems to have more decimals than the actual measurements.

Author Response

Dear Reviewer:

Thanks very much for taking your time to review our manuscript entitled “A method to predict CO2 mass concentration in sheep barn based on RF-PSO-LSTM model”(ID: animals-2233884).

We appreciate all your comments and suggestions. Those comments are all valuable and very helpful for revising and improving our paper, as well as the important guiding significance to our research. We have studied the comments carefully and have made the corrections which we hope meet with approval. We have responded to your suggestions as follows:

Point 1: This article proposed a method to predict CO2 concentration using the RF-PSO-LSTM model. The analysis seems helpful for researchers to model air quality in a similar area. I have some comments for your consideration.

Response 1: We sincerely thank you for taking the time to review our paper.

We are very grateful for your affirmation of our research results and paper, your affirmation has inspired us, and we will continue to work on it.

Point 2: I have a question regarding the need for this specific barn. Ventilation rate seems to be one of the most impactful factors in the CO2 concentration inside the barn. However, that was not included in the testing index (input variable). Sometimes it is very difficult to obtain continuous ventilation rate data, but is there a way to somehow roughly correlate ventilation rate with one of the existing testing indexes such as air temperature? Ventilation rates highly depend on temperature. As you mentioned, in summer, the barn uses natural ventilation, and in winter, it uses mechanical ventilation. If you can provide some analysis on ventilation (quantitatively or qualitatively), that would greatly enhance the application of this model.

Response 2: We are very grateful for your question. We have considered this issue carefully.

We strongly agree with you that the ventilation rate is one of the biggest factors affecting the CO2 concentration in sheep barns.

If the ventilation rate is insufficient, the air in the sheep barn will not circulate easily, carbon dioxide concentration will gradually accumulate, and the temperature in the barn will rise, leading to a decrease in air quality, which will have a negative impact on the growth and health of meat sheep. On the contrary, if the ventilation rate is too high, the air flow in the sheep barn will be too fast, which will introduce too much air and humidity, causing the temperature in the barn to drop rapidly and adversely affect the health of the meat sheep.

Therefore, the CO2 concentration and air quality in the sheep barn need to be regularly monitored, and the ventilation rate adjusted in time to ensure the health and productivity of the sheep.

We have reviewed the data and learned that there is a mathematical relationship between ventilation rate and temperature in sheep barns.

We are very sorry that we did not consider comprehensively when making the data set and did not collect the ventilation rate data of the sheep house, so we cannot provide these data temporarily. We would like to express our sincere apologies to you once again.

Your suggestion is very good and has brought great inspiration to us. We will use some simulation software (e.g. ANSYS Fluent, COMSOL Multiphysics, OpenFOAM) to simulate and analyze the airflow in the sheep barn in our next work, and then combine the collected data on ventilation rate, temperature, etc. to conduct experiments.

Once again, we would like to express our sincere gratitude to you.

Point 3: In Line 68, I would not call CO2 a “harmful gas”.

Response 3: We are very grateful for your suggestion, the CO2 cannot be called a harmful gas.

We have made changes based on your suggestions, which can be seen in the paper in the 74-75 and so on line.

Point 4: You mentioned that the height of the sensors is 2.4 and 3.0-3.1 meters from the ground. How tall is the ceiling of the barn? The barn has a decent size. Regarding your measurement equipment in the paragraph of line 172, where was the location of the equipment during sampling? How did you control the spatial variability of data with just 1 sampling location instead of multiple?

Response 4: We very much appreciate these questions that you have raised. We have thought carefully about these issues.

Here we answer:

1. We took field measurements in the sheep barn. The highest point of the top of the barn was 3.7 meters from the ground, and the lowest point of the top of the barn was 2.9 meters from the ground, and the top of the barn was sloped.

2. Our sheep barn has a beam structure at the top. When installing the equipment, the fixtures are tied vertically to the beam and then the measuring equipment is tied to the fixtures in order to carry out measurements of the environmental data of the sheep barn. In short, the measuring device is hung in the middle of the sheep barn.

3. We are very sorry. At that time, we did not consider comprehensively when collecting data, and installed a set of equipment that only carried out data collection of sheep barn at one sampling point, but not at multiple sampling points. Your suggestion is very correct. When we collect sheep barn data, we should carry out data collection of multiple sampling points and analyze the spatial variation of sheep barn data comprehensively, so that the processing can make our work more reasonable and improve the application of the model at the same time.

Our answer may not be very comprehensive and may have caused you some misunderstanding, we express our sincere apologies to you. We thank you again for your suggestions.

Point 5: The number of significant figures depends on the accuracy of the measurement equipment. The result of your model output (Table 4) seems to have more decimals than the actual measurements.

Response 5: We appreciate you asking this question. We have given this issue some careful thought.

Here we answer:

Your question is a very good one. The effective number of data we collect in the sheep shed depends on the accuracy of the measuring equipment, which is very correct.

When our model processes the collected sheep barn data, the model applies complex mathematical relationships to fit the data to enable the model to make regression predictions on the sheep barn data. The model will get data containing many decimals during this processing. In order to make the data more standardized, we unify the decimal places of the data.

Our answers may not be comprehensive, and we extend our sincere apologies to you. Once again, we thank you for your questions.

We once again express our heartfelt thanks to you.

Reviewer 3 Report

Lines 166-168 indicate that the CO2 sensors are installed at a height of 2.4 m from the ground and the rest of the sensors are 3.0-3.1 from the ground. In my opinion, the CO2 and H2S sensors should be at a height betwen 0.4 and 0.6 m from the ground so that they record the quality of the air at the height of the animals'' heads. Naturally they should be suitably protected. 

On the other hand, the temperature and humidity sensors are also placed in a very high position that can give different readings than those experienced by the animals. I believe that 0in the future trials these measures should be adjusted.

Author Response

Dear Reviewer:

Thanks very much for taking your time to review our manuscript entitled “A method to predict CO2 mass concentration in sheep barn based on RF-PSO-LSTM model”(ID: animals-2233884).

We appreciate all your comments and suggestions. Those comments are all valuable and very helpful for revising and improving our paper, as well as the important guiding significance to our research. We have studied the comments carefully and have made the corrections which we hope meet with approval. We have responded to your suggestions as follows:

Point 1: Lines 166-168 indicate that the CO2 sensors are installed at a height of 2.4 m from the ground and the rest of the sensors are 3.0-3.1 from the ground. In my opinion, the CO2 and H2S sensors should be at a height between 0.4 and 0.6 m from the ground so that they record the quality of the air at the height of the animals'' heads. Naturally they should be suitably protected.

Response 1: We are very grateful to you for these suggestions. We have given careful thought to these suggestions.

We strongly agree with you. The CO2 and H2S sensors should be installed closer to the ground so that the data we collect is similar to the air quality data breathed by the meat goats. It also allows us to better study the effect of air quality in sheep barns on meat sheep. When the sensor is installed in a low position, the daily activities of meat sheep may damage the sensor, so we need to protect the sensor.

Our answers may not be comprehensive, and we extend our sincere apologies to you. Once again, we thank you for your questions.

Point 2: On the other hand, the temperature and humidity sensors are also placed in a very high position that can give different readings than those experienced by the animals. I believe that in the future trials these measures should be adjusted.

Response 2: We are very grateful to you for this suggestion. We have given careful thought to this suggestion.

We feel that your point is very reasonable. Our temperature and humidity sensors should also be installed closer to the ground so that the data we collect is similar to the data experienced by the meat goats. At the same time, it can help us to better observe and study the effect of humidity and temperature changes on meat sheep. In the near future, we are going to go to the sheep barn to adjust the sensor installation position in order to obtain better test data.

Our answer may not be very comprehensive and we express our sincere apologies to you. We thank you again for your suggestions.

We once again express our heartfelt thanks to you.

Reviewer 4 Report

This study developed a model to predict CO2 mass concentration in sheep barn. Theoretically, I think, this developed model presented better results than other models to predict the concentration, but I don’t see any real meaning to do that for the practice. If we predict something, it means it is difficult to monitor or current methods of monitoring have some short backs. However, there are many CO2 sensors available and cheaper than those sensors of predictor used in this study, such as PM, H2S and so on. Therefore, why should we buy those kinds of instruments to predict the CO2 concentration, instead just buy CO2 monitor? I suggest the authors reduce the predicting factors and select few factors which were commonly monitored in real sheep barns.

Other specific questions:

1.     The language needs to be well polished.

2.     The section of Simple Summary describes much on the background and has less information on the outputs of this study.

3.     The abstract needs give the information of factors used to predict the CO2 concentration.

4.     Line 165-170, what are the basis for the setting of those sensors?

5.     Explain the “Agreement” in Table 1 with note or other forms.

6.     Line 256-276, it need to well explain the symbols of those parameters presented, what is Ct-1, ft etc.?

7.     Some of information listed in the section of results and discussion should be put in the section of methods, such as Line 357-361.

8.     Line 401, what does it means that the R2 increased by 0.006?

9.     How was the “feature importance score” in Table 6 calculated? It should be explained in methods. The results should be well explained as well in the abstract or conclusion. Will it be enough to use the firs three parameters to predict? I also suggest to add some comparison of results with those from other research.

Author Response

Dear Reviewer:

Thanks very much for taking your time to review our manuscript entitled “A method to predict CO2 mass concentration in sheep barn based on RF-PSO-LSTM model”(ID: animals-2233884).

We appreciate all your comments and suggestions. Those comments are all valuable and very helpful for revising and improving our paper, as well as the important guiding significance to our research. We have studied the comments carefully and have made the corrections which we hope meet with approval. We have responded to your suggestions as follows:

Point 1: This study developed a model to predict CO2 mass concentration in sheep barn. Theoretically, I think, this developed model presented better results than other models to predict the concentration, but I don’t see any real meaning to do that for the practice. If we predict something, it means it is difficult to monitor or current methods of monitoring have some short backs. However, there are many CO2 sensors available and cheaper than those sensors of predictor used in this study, such as PM, H2S and so on. Therefore, why should we buy those kinds of instruments to predict the CO2 concentration, instead just buy CO2 monitor? I suggest the authors reduce the predicting factors and select few factors which were commonly monitored in real sheep barns.

Response 1: We sincerely thank you for taking the time to review our paper. We are very grateful for your affirmation of our research results and paper, your affirmation has inspired us, and we will continue to work on it.

We very much appreciate these questions that you have raised. We have thought carefully about these issues.

We agree with your point of view. If we make a prediction about something, it means that it is difficult to be monitored or that there are some flaws in the current monitoring methods, which is very true.

We do sheep barn CO2 concentration prediction because when sheep barn accumulates CO2 in excess, it will lead to chronic hypoxia, loss of appetite, weakness and stress, which seriously endangers the healthy growth of sheep.

Our model can predict the trend of CO2 concentration in sheep barn in advance. When the predicted CO2 concentration value is outside the normal range, our model will send corresponding commands to the system, which will intelligently regulate the sheep barn environment by adjusting the equipment in the barn (e.g., ventilation fans, windows and doors, etc.). This operation allows the CO2 concentration in the sheep barn to remain within the normal range, reducing the damage to the growth of meat sheep caused by high CO2 concentrations while improving the welfare of the meat sheep.

The sheep barn CO2 concentration prediction we did is part of the sheep barn environmental intelligent control system, which is still in the process of commissioning. We will continue to work hard to apply it in the sheep barn soon and increase its practical value.

Our answers may not be very comprehensive and may have caused you some misunderstandings, and we express our sincere apologies to you. We thank you again for your suggestions.

Other specific questions:

Point 2: The language needs to be well polished.

Response 2: We are very grateful to you for this suggestion. We polished the language of the paper based on your suggestions and asked a native speaker to check the essay. We thank you very sincerely for your help with the paper.

Point 3: The section of Simple Summary describes much on the background and has less information on the outputs of this study.

Response 3: We are very grateful to you for this suggestion. We have added information on the results of this study in the Simple Summary section based on your suggestions. This suggestion of yours can make our Simple Summary part become more reasonable. Once again, we would like to express our heartfelt thanks to you. Which can be seen in line 27-33 of the paper.

Point 4: The abstract needs give the information of factors used to predict the CO2 concentration.

Response 4: We are very grateful to you for this suggestion. We have added information on the factors used to predict CO2 concentration to the abstract section based on your suggestion, which can be seen in line 45-46 of the paper. This suggestion of yours can make our abstract part more complete and reasonable. Once again, we would like to express our heartfelt thanks to you.

Point 5: Line 165-170, what are the basis for the setting of those sensors?

Response 5: We appreciate you asking this question. We have given this issue some careful thought.

Here we answer:

We took field measurements in the sheep barn. The highest point of the top of the barn was 3.7 meters from the ground, and the lowest point of the top of the barn was 2.9 meters from the ground, and the top of the barn was sloped.

Our sheep barn has a beam structure at the top. When installing the equipment, the fixtures are tied vertically to the beam and then the measuring equipment is tied to the fixtures in order to carry out measurements of the environmental data of the sheep barn. In short, the measuring device is hung in the middle of the sheep barn.

We extend our sincere apologies to you. We did not consider comprehensively when installing sensor equipment. The CO2 and H2S sensors should be installed closer to the ground so that the data we collect is similar to the air quality data breathed by the meat goats. It also allows us to better study the effect of air quality in sheep barns on meat sheep.

Our temperature and humidity sensors should also be installed closer to the ground so that the data we collect is similar to the data experienced by the meat goats. At the same time, it can help us to better observe and study the effect of humidity and temperature changes on meat sheep. In the near future, we are going to go to the sheep barn to adjust the sensor installation position in order to obtain better test data.

Our answers may not be very comprehensive and may have caused you some misunderstandings, and we express our sincere apologies to you. We thank you again for your suggestions.

Point 6: Explain the “Agreement” in Table 1 with note or other forms.

Response 6: We are very grateful to you for this suggestion. We have explained "Agreement" in accordance with your suggestion, which can be seen in line 189-197 of the paper. Your suggestion can make our article more complete. Once again, we would like to express our heartfelt thanks to you.

Point 7: Line 256-276, it need to well explain the symbols of those parameters presented, what is Ct-1, ft etc.?

Response 7: We are very grateful to you for this suggestion. We have detailed the parameters according to your suggestions, which can be seen in line 269-294 of the paper. Your suggestion can make our article more reasonable. Once again, we would like to express our heartfelt thanks to you.

Point 8: Some of information listed in the section of results and discussion should be put in the section of methods, such as Line 357-361.

Response 8: We are very grateful to you for this suggestion. We have made changes based on your suggestions, which can be seen in line 376-389 of the paper. Your suggestion can make the structure of our article more standardized. Once again, we thank you.

Point 9: Line 401, what does it means that the R2 increased by 0.006?

Response 9: We are very grateful to you for raising this issue. We have given this issue some careful thought.

Here we answer:

The coefficient of determination (R2) is a common measure of how well a regression model fits the actual data. The value of R2 ranges from 0 to 1. The closer the value of R2 is to 1, the better the fit of the model to the actual data, and vice versa.

In the paper, the R2 value of the two hidden layer model increased by 0.006 compared to the R2 value of the one hidden layer model.

Although the increase is not significant, it is clear from the definition of R2 that the two hidden layer model performs better than the one hidden layer model in this experiment.

Our answers may not be very comprehensive and may have caused you some misunderstandings, and we express our sincere apologies to you. We thank you again for your suggestions.

Point 10: How was the “feature importance score” in Table 6 calculated? It should be explained in methods. The results should be well explained as well in the abstract or conclusion. Will it be enough to use the firs three parameters to predict? I also suggest to add some comparison of results with those from other research.

Response 10: We are very grateful to you for these questions and suggestions. We have given these questions and suggestions careful thought.

Here we answer:

The Random Forest algorithm is an integrated decision tree based learning method that calculates feature importance scores based on Out-Of-Bag (OOB) error rates and tree structure.

Specifically, the random forest algorithm chooses a random portion of features to split as it constructs each tree. This creates some features that are not used by the tree. When using random forests for prediction, feature importance can be calculated by using these unused features.

When constructing a random forest, each tree is trained based on a portion of the data, while the rest of the data is not used for training. This data that is not used for training is called OOB data.

For each feature, an importance score can be calculated by evaluating the performance of the random forest on the OOB data. Specifically, the feature importance score can be calculated as follows:

  1. for each feature, calculate the OOB error for that feature on each tree of the random forest. The OOB error is the prediction error calculated for each OOB sample on that tree.
  2. Calculate the mean error of all OOB samples. This average error can be considered as the importance score of the feature.
  3. Repeat the above steps for all features to obtain the final importance score for each feature.

In summary, this is the approximate flow of the Random Forest algorithm for calculating feature importance scores.

We provide a brief description in the "2.2.1. Random Forests feature importance ranking" section of the paper, which can be found in line 237-251 of the paper. It is also briefly described in the " Abstract " and the " Conclusion ".

The first four parameters (light intensity, air relative humidity, air temperature, and PM2.5 mass concentration) were chosen as inputs to the model after a comprehensive comparison test in our study. Our specific tests and analyses can be seen in the 494-517 lines of the paper.

Our answers may not be very comprehensive and may have caused you some misunderstandings, and we express our sincere apologies to you.

The suggestions and questions you have raised are all very good, which can make our paper more reasonable and standardized.

We thank you again for these suggestions and questions.

We once again express our heartfelt thanks to you.

Round 2

Reviewer 4 Report

Some of the answers to the questions were not corrected or explained in the main text.

Author Response

Dear Reviewer:

Thanks very much for taking your time to review our manuscript entitled “A method to predict CO2 mass concentration in sheep barn based on RF-PSO-LSTM model”(ID: animals-2233884).

We appreciate all your comments and suggestions. Those comments are all valuable and very helpful for revising and improving our paper, as well as the important guiding significance to our research. We have studied the comments carefully and have made the corrections which we hope meet with approval. We have responded to your suggestions as follows:

Point 1: Some of the answers to the questions were not corrected or explained in the main text.

Response 1: We express our sincere apologies to you. We strongly agree with your suggestion to make extensive language revisions to the paper.

Our language modification ability is insufficient. During the first modification, we only discovered a few errors and made simple language modifications. We once again express our sincere apologies to you.

We strongly agree with your suggestion this time. We have chosen your organization's language modification service (MDPI English Editing Service) to address our lack of language modification capabilities.

The language polishing service is very effective. After polishing, our paper has become more standardized in language expression, more logical at the language level, and improved the reading experience of readers.

We thank you very sincerely for your help with the paper.

Our answers may not be very comprehensive and may have caused you some misunderstandings, and we express our sincere apologies to you. We thank you again for your suggestions.

We once again express our heartfelt thanks to you.